# Surface-to-volume scaling and aspect ratio preservation in rod-shaped bacteria

Nikola Ojkic, Diana Serbanescu, Shiladitya Banerjee[†]*

Department of Physics and Astronomy, Institute for the Physics of Living Systems, University College London, London, United Kingdom

**Abstract** Rod-shaped bacterial cells can readily adapt their lengths and widths in response to environmental changes. While many recent studies have focused on the mechanisms underlying bacterial cell size control, it remains largely unknown how the coupling between cell length and width results in robust control of rod-like bacterial shapes. In this study we uncover a conserved surface-to-volume scaling relation in *Escherichia coli* and other rod-shaped bacteria, resulting from the preservation of cell aspect ratio. To explain the mechanistic origin of aspect-ratio control, we propose a quantitative model for the coupling between bacterial cell elongation and the accumulation of an essential division protein, FtsZ. This model reveals a mechanism for why bacterial aspect ratio is independent of cell size and growth conditions, and predicts cell morphological changes in response to nutrient perturbations, antibiotics, MreB or FtsZ depletion, in quantitative agreement with experimental data.
DOI: https://doi.org/10.7554/eLife.47033.001

*For correspondence:
shiladitya.banerjee@ucl.ac.uk

Present address: [†]Department of Physics, Carnegie Mellon University, Pittsburgh, United States

Competing interests: The authors declare that no competing interests exist.

## Introduction

Cell morphology is an important adaptive trait that is crucial for bacterial growth, motility, nutrient uptake, and proliferation (*Young, 2006*). When rod-shaped bacteria grow in media with different nutrient availability, both cell length and width increase with growth rate (*Schaechter et al., 1958*; *Si et al., 2017*). At the single-cell level, control of cell volume in many rod-shaped cells is achieved via an *adder* mechanism, whereby cells elongate by a fixed length per division cycle (*Amir, 2014*; *Campos et al., 2014*; *Taheri-Araghi et al., 2015*; *Deforet et al., 2015*; *Wallden et al., 2016*; *Banerjee et al., 2017*). A recent study has linked the determination of cell size to a condition-dependent regulation of cell surface-to-volume ratio (*Harris and Theriot, 2016*). However, it remains largely unknown how cell length and width are coupled to regulate rod-like bacterial shapes in diverse growth conditions (*Volkmer and Heinemann, 2011*; *Belgrave and Wolgemuth, 2013*; *Colavin et al., 2018*; *Shi et al., 2018*).

## Results

Here we investigated the relation between cell surface area ($S$) and cell volume ($V$) for *E. coli* cells grown under different nutrient conditions, challenged with antibiotics, protein overexpression or depletion, and single gene deletions (*Nonejuie et al., 2013*; *Harris and Theriot, 2016*; *Si et al., 2017*; *Vadia et al., 2017*; *Campos et al., 2018*; *Gray et al., 2019*). Collected surface and volume data span two orders of magnitude and exhibit a single power law in this regime: $S = \gamma V^{2/3}$ (*Figure 1A*). Specifically, during steady-state growth (*Si et al., 2017*), $\gamma = 6.24 \pm 0.04$, suggesting an elegant geometric relation: $S \approx 2\pi V^{2/3}$. This surface-to-volume scaling with a constant prefactor, $\gamma$, is a consequence of tight control of cell aspect ratio $\eta$ (length/width) (*Figure 1D*), whose mechanistic origin has been puzzling for almost half a century (*Zaritsky, 1975*; *Zaritsky, 2015*). Specifically, for a

sphero-cylindrical bacterium, $S = \gamma V^{2/3}$ implies $\gamma = \eta\pi\left(\frac{\eta\pi}{4} - \frac{\pi}{12}\right)^{-2/3}$. A constant $\gamma$ thus defines a constant aspect ratio $\eta = 4.14 \pm 0.17$ (**Figure 1B**-inset), with a coefficient of variation ~14% (**Figure 1B**).

The surface-to-volume relation for steady-state growth, $S \approx 2\pi V^{2/3}$, results in a simple expression for cell surface-to-volume ratio: $S/V \approx 2\pi V^{-1/3}$. Using the phenomenological nutrient growth law $V = V_0 e^{\alpha\kappa}$ (**Schaechter et al., 1958**), where $\kappa$ is the population growth rate, a negative correlation emerges between $S/V$ and $\kappa$:

$$S/V \approx 2\pi V_0^{-1/3} e^{-\alpha\kappa/3} , \tag{1}$$

with $V_0$ the cell volume at $\kappa = 0$, and $\alpha$ is the relative rate of increase in $V$ with $\kappa$ (**Figure 1C**). In **Equation (1)** underlies an adaptive feedback response of the cell — at low nutrient concentrations, cells increase their surface-to-volume ratio to promote nutrient influx (**Si et al., 2017**; **Harris and Theriot, 2018**). Prediction from **Equation (1)** is in excellent agreement with the best fit to the experimental data. Furthermore, a constant aspect ratio of $\approx 4$ implies $V \approx \sqrt{8}w^3$ and $S \approx 4\pi w^2$, where $w$ is the cell width, suggesting stronger geometric constraints than recently proposed (**Harris and Theriot, 2018**; **Shi et al., 2018**). Thus, knowing cell volume as a function of cell cycle parameters (**Si et al., 2017**) we can directly predict cell width and length under changes in growth media, in agreement with experimental data (**Figure 1—figure supplement 1A–B**). We further analysed cell shape data for 48 rod-shaped bacteria, one rod-shaped Archaea (*H. vulcanii*), two long spiral Spirochete, and one coccoid bacteria (**Figure 1E**). Collected data for all rod-shaped cells follow closely the relationship $S \approx 2\pi V^{2/3}$, while the long Spirochetes deviate from this curve (**Figure 1D–E**). Coccoid *S. aureus* also follows the universal scaling relation $S = \gamma V^{2/3}$ (with $\gamma = 4.92$), but maintains a much lower aspect ratio $\eta = 1.38 \pm 0.18$ (**Quach et al., 2016**) when exposed to different antibiotics (**Figure 1D–E**). Therefore, aspect-ratio preservation likely emerges from a mechanism that is common to diverse rod-shaped and coccoid bacterial species.

To investigate how aspect ratio is regulated at the single cell level we analysed the morphologies of *E. coli* cells grown in the mother machine (**Taheri-Araghi et al., 2015**) (**Figure 2A,B**). For five different growth media, mean volume and surface area of newborn cells also follow the relationship $S = 2\pi V^{2/3}$, suggesting that a fixed aspect ratio is maintained on average. In the single-cell data, slight deviation from the 2/3 scaling is a consequence of large fluctuations in newborn cell lengths for a given cell width (**Figure 2—figure supplement 1A–B**). Importantly, the probability distribution of aspect ratio is independent of the growth media (**Figure 2B**), implying that cellular aspect ratio is independent of cell size as well as growth rate.

To explain the origin of aspect ratio homeostasis we developed a quantitative model for cell shape dynamics that accounts for the coupling between cell elongation and the accumulation of cell division proteins FtsZ (**Figure 2C**). Our model is thus only applicable to bacteria that divide using the FtsZ machinery. *E. coli* and other rod-like bacteria maintain a constant width during their cell cycle while elongating exponentially in length $L$ (**Donachie et al., 1976**; **Taheri-Araghi et al., 2015**): $\mathrm{d}L/\mathrm{d}t = kL$, with $k$ the elongation rate. Cell division is triggered when a constant length is added per division cycle — a mechanism that is captured by a model for threshold accumulation of division initiator proteins, produced at a rate proportional to cell size (**Basan et al., 2015**; **Deforet et al., 2015**; **Ghusinga et al., 2016**). While many molecular candidates have been suggested as initiators of division (**Adams and Errington, 2009**), a recent study (**Si et al., 2019**) has identified FtsZ as the key initiator protein that assembles a ring-like structure in the mid-cell region to trigger septation.

Dynamics of division protein accumulation can be described using a two-component model. First, a cytoplasmic component with abundance $P_c$ grows in proportion to cell size ($\propto L$), as ribosome content increases with cell size (**Marguerat and Bähler, 2012**). Second, a ring-bound component, $P_r$, is assembled from the cytoplasmic pool at a constant rate. Dynamics of the cytoplasmic and ring-bound FtsZ are given by:

$$\frac{\mathrm{d}P_c}{\mathrm{d}t} = -k_b P_c + k_d P_r + k_P L , \tag{2}$$

$$\frac{\mathrm{d}P_r}{\mathrm{d}t} = k_b P_c - k_d P_r , \tag{3}$$

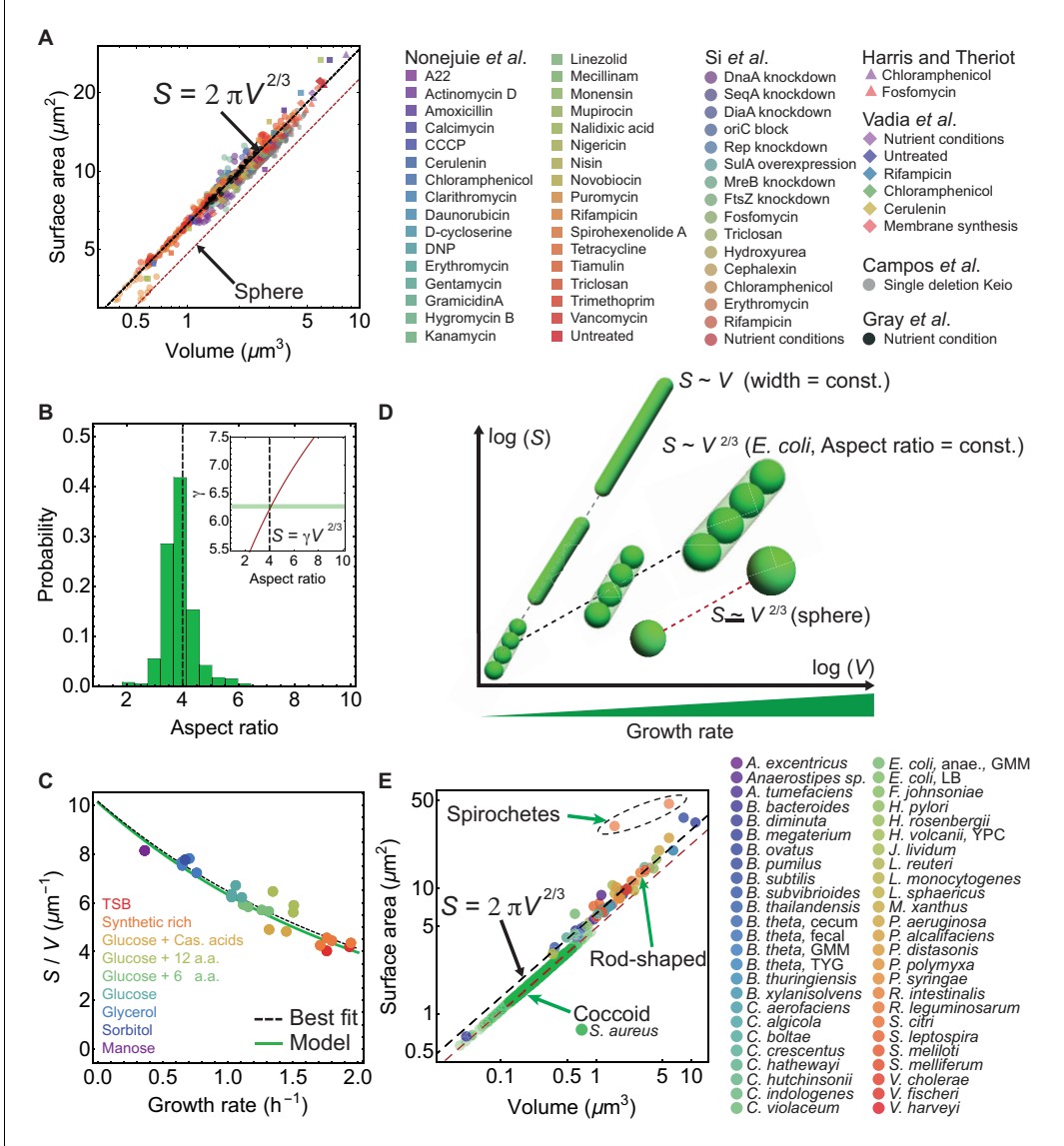

**Figure 1.** Surface-to-volume scaling in *E. coli* and other rod-shaped bacteria. (A) *E. coli* cells subjected to different antibiotics, nutrient conditions, protein overexpression/depletion, and single gene deletions (*Nonejuie et al., 2013*; *Si et al., 2017*; *Harris and Theriot, 2016*; *Vadia et al., 2017*; *Campos et al., 2018*; *Gray et al., 2019*), follow the scaling relation between population-averaged surface area ($S$) and volume ($V$): $S = \gamma V^{2/3}$ (legend on the right, 5011 data points; *Supplementary file 1*). Best fit shown in dashed black line for steady-state data from *Si et al. (2017)* gives $\gamma = 6.24 \pm 0.04$, and a power law exponent $0.671 \pm 0.006$. For single deletion Keio set (*Campos et al., 2018*), the best fit curve is $S = 5.79\,V^{2/3}$. (B) Aspect-ratio distribution for cells growing in steady-state, corresponding to the data in (A) (*Si et al., 2017*). (Inset) Relationship between $\gamma$ and aspect ratio $\eta$ for a sphero-cylinder (red line). Best fit from (A) shown with horizontal green band gives aspect ratio $4.14 \pm 0.17$. (C) $S/V$ vs growth rate. Model line uses $S = 2\pi V^{2/3}$ and the nutrient growth law (*Equation 1*). Data from *Si et al. (2017)*. (D) $S$-$V$ relation for various bacterial cell shapes. Black dashed line: Small, medium, and large rod-shaped cells with a conserved aspect ratio of 4 follow the relation: $S = 2\pi V^{2/3}$. Gray dashed line: Filamentous cells with constant cell width follow the scaling law: $S \sim V$. Red dashed line: Spheres follow $S = 6^{2/3}\pi^{1/3}V^{2/3}$. (E) $S$ vs $V$ for 49 different bacterial species (*Sato, 2000*; *Trachtenberg, 2004*; *Pelling et al., 2005*; *Wright et al., 2015*; *Deforet et al., 2015*; *Desmarais et al., 2015*; *Harris and Theriot, 2016*; *Ojkic et al., 2016*; *Quach et al., 2016*; *Carabetta et al., 2016*; *Chattopadhyay et al., 2017*; *Lopez-Garrido et al., 2018*; *Gray et al., 2019*), and one rod-shaped Archaea (*H. volcanii*) (*Supplementary file 2*). Rod-shaped cells lie on $S = 2\pi V^{2/3}$ line, above the line are Spirochete and below the line are coccoid. For coccoid *S. aureus* exposed to different antibiotics best fit is $S = 4.92\,V^{2/3}$, with preserved aspect ratio $\eta = 1.38 \pm 0.18$. Red dashed line is for spheres.

DOI: https://doi.org/10.7554/eLife.47033.002

The following figure supplement is available for figure 1:

**Figure supplement 1.** Control of cell width and length in *E. coli*.

*Figure 1 continued on next page*

*Figure 1 continued*

DOI: https://doi.org/10.7554/eLife.47033.003

where $k_P$ is the constant production rate of cytoplasmic FtsZ, $k_b$ is the rate of binding of cytoplasmic FtsZ to the Z-ring, and $k_d$ is the rate of disassembly of Z-ring bound FtsZ. At the start of the cell cycle, we have $P_c = P^*$ (a constant) and $P_r = 0$. Cell divides when $P_r$ reaches a threshold amount, $P_0$, required for the completion of ring assembly. A key ingredient of our model is that $P_0$ scales linearly with the cell circumference, $P_0 = \rho\pi w$, preserving the density $\rho$ of FtsZ in the ring. This is consistent with experimental findings that the total FtsZ scales with the cell width (*Shi et al., 2017*). Accumulation of FtsZ proteins, $P = P_c + P_r - P^*$, follows the equation: $\mathrm{d}P/\mathrm{d}t = k_P L$, where $k_P$ is the production rate of division proteins, with $P = 0$ at the start of the division cycle. We assume that $k_b \gg k_d$, such that all the newly synthesized cytoplasmic proteins are recruited to the Z-ring at a rate much faster than growth rate (*Söderström et al., 2018*). As a result, cell division occurs when $P = P_0$ (*Figure 2C*). Upon division $P$ is reset to 0 for the two daughter cells. It is reasonable to assume that all the FtsZ proteins are in filamentous form at cell division, as the concentration of FtsZ in an average *E. coli* cell is in the range 4–10 µM, much higher than the critical concentration 1 µM (*Erickson et al., 2010*).

From the model it follows that during one division cycle cells grow by adding a length $\Delta L = P_0 k/k_P$, which equals the homeostatic length of newborn cells. Furthermore, recent experiments suggest that the amount of FtsZ synthesised per unit cell length, $\mathrm{d}P/\mathrm{d}L$, is constant (*Si et al., 2019*). This implies,

$$\frac{\mathrm{d}L}{\mathrm{d}P} = \frac{k}{k_P} = \frac{\Delta L}{P_0} \propto \frac{\Delta L}{w} = const. \tag{4}$$

Aspect ratio homeostasis is thus achieved via a balance between the rates of cell elongation and division protein production, consistent with observations that FtsZ overexpression leads to minicells and FtsZ depletion induces elongated phenotypes (*Potluri et al., 2012*; *Zheng et al., 2016*). Indeed single cell *E. coli* data (*Taheri-Araghi et al., 2015*) show that $\Delta L/w$ is constant on average and independent of growth conditions (*Figure 2D*). Furthermore, added length correlates with cell width during one cell cycle implying that the cell width is a good predictor for added cell length (*Figure 2—figure supplement 1C–D*).

To predict cell-shape dynamics under perturbations to growth conditions we simulated our single-cell model (*Figure 3*, Materials and methods) with an additional equation for cell width that we derived from a recent model proposed by *Harris and Theriot (2016)*: $\mathrm{d}S/\mathrm{d}t = \beta V$, where $\beta$ is the rate of surface area synthesis relative to volume and is a linearly increasing function of $k$ (*Figure 3—figure supplement 1A*). This model leads to an equation for the control of cell width for a spherocylinder shaped bacterium,

$$\frac{\mathrm{d}w}{\mathrm{d}t} = w(k - \beta w/4)\frac{1 - w/3L}{1 - w/L}, \tag{5}$$

such that $w = 4k/\beta$ at steady-state. It then follows from *Equation (4)* that the added cell length $\Delta L \propto k^2/\beta k_P$. However, our model for division control is mechanistically different from *Harris and Theriot (2016)*. In the latter, cells accumulate a threshold amount of excess surface area material to trigger septation, which does not lead to aspect ratio preservation. By contrast, we propose that cells divide when they accumulate a threshold amount of division proteins in the Z-ring, proportional to the cell diameter.

We simulated nutrient shift experiments using the coupled equations for cell length, width and division protein production (Materials and methods). When simulated cells are exposed to new nutrient conditions (*Figure 3—figure supplement 1B–E*), changes in cell width result in a transient increase in aspect ratio ($\eta = L/w$) during nutrient downshift, or a transient decrease in $\eta$ during nutrient upshift (*Figure 3C*). After nutrient shift, aspect ratio reaches its pre-stimulus homeostatic value over multiple generations. Typical timescale for transition to the new steady-state is controlled by the growth rate of the new medium ($\propto k^{-1}$), such that the cell shape parameters reach a steady state faster in media with higher growth rate. This result is consistent with the experimental observation that newborn aspect ratio reaches equilibrium faster in fast growing media (*Taheri-Araghi et al.,*

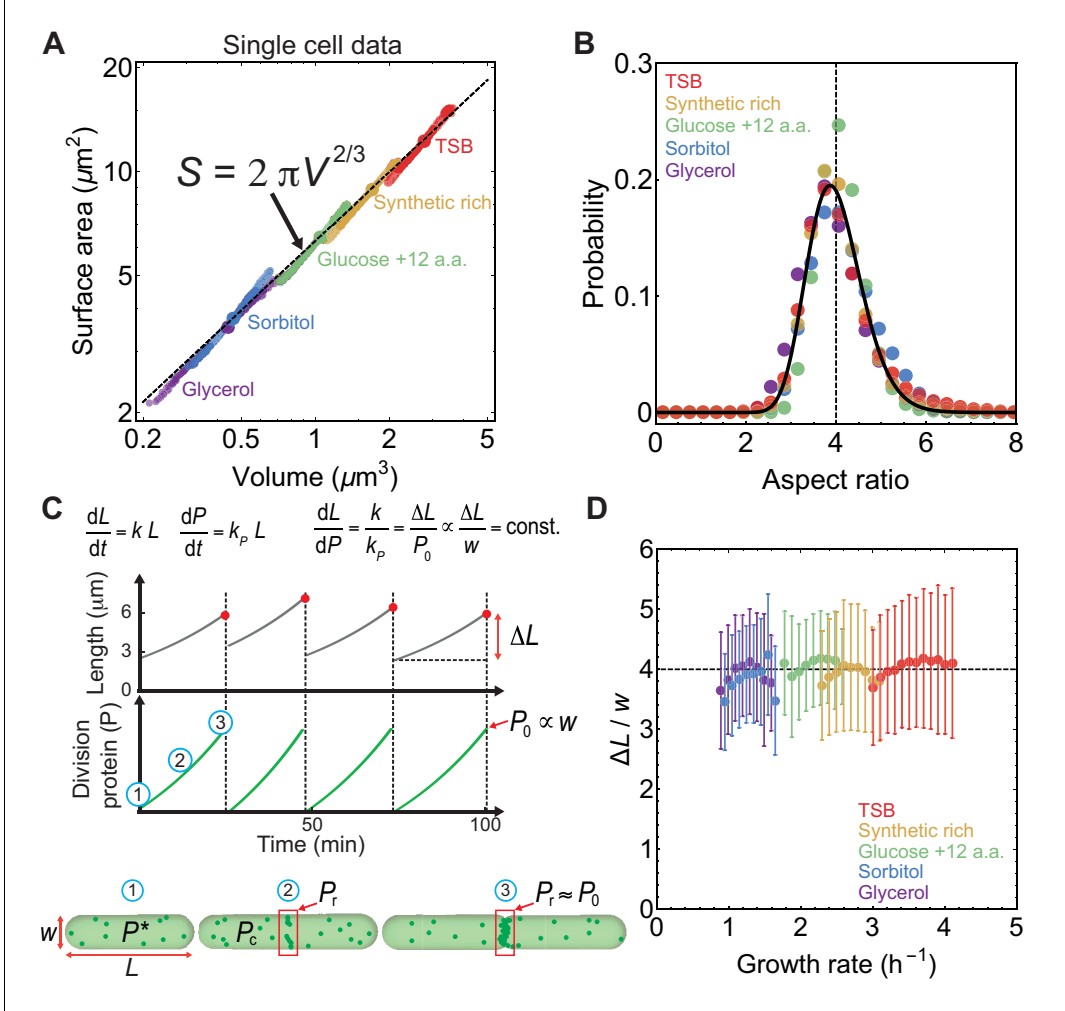

**Figure 2.** Aspect ratio control in *E. coli* at the single cell level. (**A**) $S$ vs $V$ for newborn *E. coli* cells grown in mother machine (*Taheri-Araghi et al., 2015*). Single cell data (small circles) binned in volume follow population averages (large circles). For sample size refer to *Supplementary file 1*. (**B**) Probability distribution of newborn cell aspect ratio is independent of growth rate, fitted by a log-normal distribution (solid line) (**C**) Model schematic. Cell length $L$ increases exponentially during the division cycle at a rate $k$. Division proteins ($P$) are produced at a rate $k_P$, and assembles a ring in the mid-cell region. At birth, cells contain $P^*$ molecules in the cytoplasm. Amount of FtsZ recruited in the ring is $P_r$. Cells divide when $P_r = P_0 \propto w$, where $w$ is cell width. $P$ vs time and $L$ vs time are reproduced from model simulations. (**D**) Ratio of the added length ($\Delta L$) and cell width ($w$) during one cell cycle is constant and independent of growth rate. Error bars: ±1 standard deviation.

DOI: https://doi.org/10.7554/eLife.47033.004

The following figure supplement is available for figure 2:

**Figure supplement 1.** Deviation from average surface-to-volume scaling law, and correlation between added length and width from single-cell *E. coli* data.

DOI: https://doi.org/10.7554/eLife.47033.005

2015) (*Figure 3—figure supplement 1F*). In our model, cell shape changes are controlled by two parameters: the ratio $k/k_P$ that determines cell aspect ratio, and $k/\beta$ that controls cell width (*Figure 4A*). Nutrient upshift or downshift only changes the ratio $k/\beta$ while keeping the steady-state aspect ratio ($\propto k/k_P$) constant.

We further used our model to predict drastic shape changes leading to deviations from the homeostatic aspect ratio when cells are perturbed by FtsZ knockdown, MreB depletion, and antibiotic treatments that induce non steady state filamentation (*Figure 4B*). First, FtsZ depletion results in long cells while the width stays approximately constant, $S \propto V^{0.95}$ (*Figure 4—figure supplement 1C*), data from *Zheng et al. (2016)*. We modelled FtsZ knockdown by decreasing $k_P$ and simulations

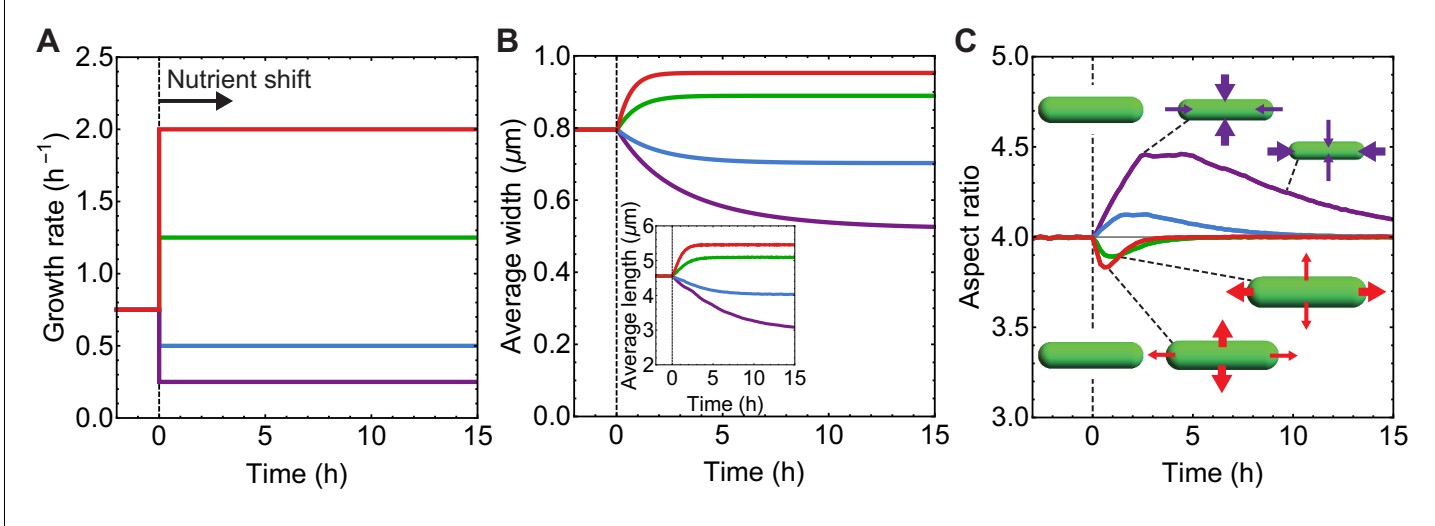

**Figure 3.** Simulations of aspect ratio preservation during nutrient upshift or downshift. (**A–C**) At t = 0 h cells are exposed to nutrient upshift or downshift. Population average of n = 10⁵ simulated cells. (**A**) Growth rate (*k*) vs time used as input for our simulations. (**B**) Population-averaged cell length and width vs time. (**C**) Population-averaged aspect ratio of newborn cells vs time. Changes in cell width and length result in a transient increase in aspect ratio during nutrient downshift, or a transient decrease during nutrient upshift.
DOI: https://doi.org/10.7554/eLife.47033.006

The following figure supplement is available for figure 3:

**Figure supplement 1.** Simulations of nutrient upshift and aspect ratio equilibration at the single cell level.
DOI: https://doi.org/10.7554/eLife.47033.007

quantitatively agree with experimental data. Second, MreB depletion increases the cell width and slightly decreases cell length while keeping growth rate constant (*Zheng et al., 2016*). We modelled MreB knockdown by decreasing $\beta$ as expected for disruption in cell wall synthesis machinery, while simultaneously increasing $k_P$ (Materials and methods). This increase in $k_P$ is consistent with a prior finding that in MreB mutant cells of various sizes, the total FtsZ scales with the cell width (*Shi et al., 2017*). Furthermore, cells treated with MreB inhibitor A22 induce envelope stress response system (Rcs) that in turn activates FtsZ overproduction (*Carballès et al., 1999*; *Cho et al., 2014*). Third, transient long filamentous cells result from exposure to high dosages of cell-wall targeting antibiotics that prevent cell division, or DNA-targeting antibiotics that induce filamentation via SOS response (*Nonejuie et al., 2013*). Cell-wall targeting antibiotics inhibit the activity of essential septum forming penicillin binding proteins, preventing cell septation. We modelled this response as an effective reduction in $k_P$, while slightly decreasing surface synthesis rate $\beta$ (Materials and methods). For DNA targeting antibiotics, FtsZ is directly sequestered during SOS response resulting in delayed ring formation and septation (*Chen et al., 2012*). Surprisingly all filamentous cells have a similar aspect ratio of 11.0 ± 1.4, represented by a single curve in the *S-V* plane (*Figure 4B*).

## Discussion

The conserved surface-to-volume scaling in diverse bacterial species, $S \sim V^{2/3}$, is a direct consequence of aspect-ratio homeostasis at the single-cell level. We present a regulatory model (*Figure 2C*) where aspect-ratio control is the consequence of a constant ratio between the rate of cell elongation (*k*) and division protein accumulation ($k_P$). Deviation from the homeostatic aspect ratio is a consequence of altered $k/k_P$, as observed in filamentous cells, FtsZ or MreB depleted cells (*Figure 4B*). By contrast, drugs that target cell wall biogenesis, for example Fosfomycin, do not alter $k/k_P$ and maintain cellular aspect ratio (*Figure 4—figure supplement 1C*).

Our study suggests that cell width is an essential shape parameter for determining cell length in *E. coli* (*Figure 2—figure supplement 1C–D*). This is to be contrasted with *B. subtilis*, where cell width stays approximately constant across different media, while elongating in length (*Sharpe et al.,*

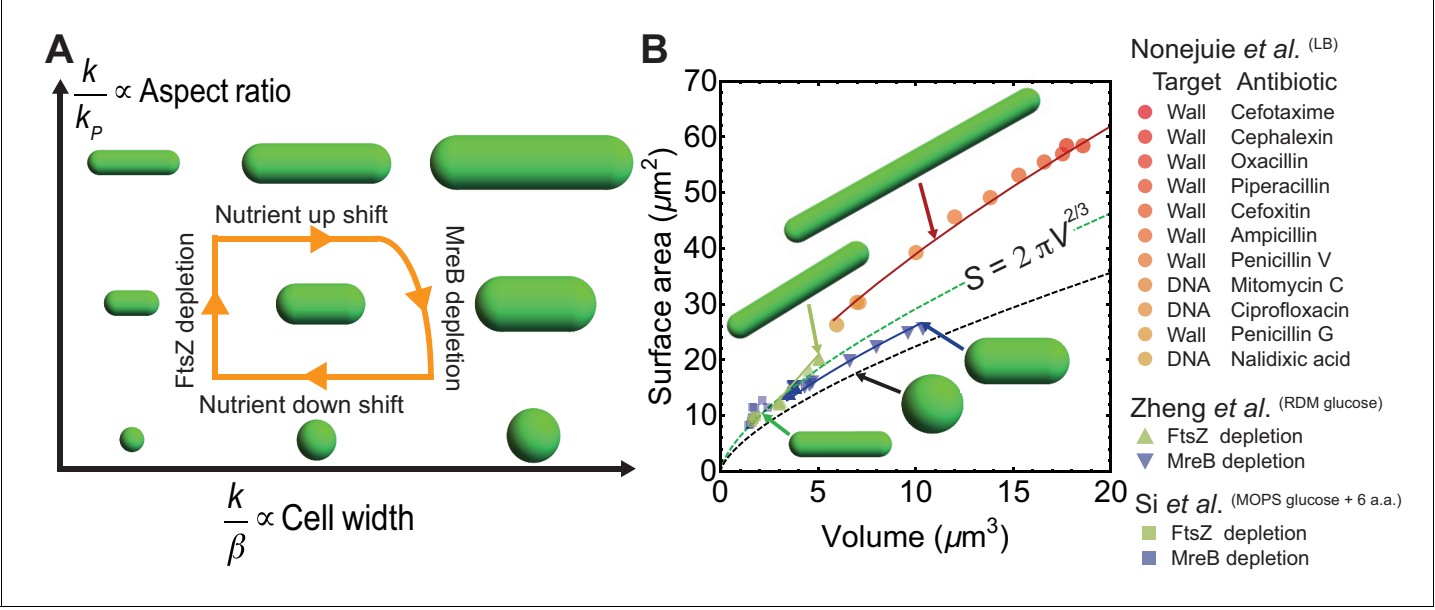

**Figure 4.** Model predictions for aspect ratio and shape control under perturbations. (A) Model parameters that control changes in cell aspect ratio ($k/k_P$) or width ($k/\beta$). For quantification see *Figure 4—figure supplement 1A*. (B) Surface area vs volume for cells under antibiotic treatment (*Nonejuie et al., 2013*), FtsZ knockdown and MreB depletion (*Si et al., 2017*; *Zheng et al., 2016*). Solid lines are best fit obtained using our model and data from *Zheng et al. (2016)* (see Materials and methods). Cells with depleted FtsZ have elongated phenotypes, while depleted MreB have smaller aspect ratio and larger width. Cell wall or DNA targeting antibiotics induce filamentation. Dashed green line: $S = 2\pi V^{2/3}$, dashed black line: spheres.

DOI: https://doi.org/10.7554/eLife.47033.008

The following figure supplement is available for figure 4:

**Figure supplement 1.** Impact of surface-to-volume ratio and FtsZ depletion on cell aspect ratio.
DOI: https://doi.org/10.7554/eLife.47033.009

*1998*). However, FtsZ recruitment in *B. subtilis* is additionally controlled by effector UgtP, which localises to the division site in a nutrient-dependent manner and prevents Z-ring assembly (*Weart et al., 2007*). This can be interpreted as a reduction in $k_P$ with increasing $k$, within the framework of our model. As a result, *B. subtilis* aspect ratio ($\propto k/k_P$) is predicted to increase with increasing growth rate.

Aspect ratio control may have several adaptive benefits. For instance, increasing cell surface-to-volume ratio under low nutrient conditions can result in an increased nutrient influx to promote cell growth (*Figure 1C*). Under translation inhibition by ribosome-targeting antibiotics, bacterial cells increase their volume while preserving aspect ratio (*Harris and Theriot, 2016*; *Si et al., 2017*). This leads to a reduction in surface-to-volume ratio to counter further antibiotic influx. Furthermore, recent studies have shown that the efficiency of swarming bacteria strongly depends on their aspect ratio (*Ilkanaiv et al., 2017*; *Jeckel et al., 2019*). The highest foraging speed has been observed for aspect ratios in the range 4–6 (*Ilkanaiv et al., 2017*), suggesting that the maintenance of an optimal aspect ratio may have evolutionary benefits for cell swarmers.

## Materials and methods

### Cell shape analysis

Bacterial cell surface area and volume are obtained directly from previous publications where these values were reported (*Si et al., 2017*; *Campos et al., 2018*; *Gray et al., 2019*), or they are calculated assuming a sphero-cylindrical cell geometry using reported values for population-averaged cell length and width (*Vadia et al., 2017*; *Nonejuie et al., 2013*; *Zheng et al., 2016*; *Wright et al., 2015*; *Deforet et al., 2015*; *Desmarais et al., 2015*; *Ojkic et al., 2016*; *Carabetta et al., 2016*; *Lopez-Garrido et al., 2018*). Single cell data are obtained from Suckjoon Jun lab (UCSD) (*Taheri-*

*Araghi et al., 2015*). For number of cells analyzed per growth condition see *Supplementary file 1*. Intergeneration autocorrelation function (*Figure 2—figure supplement 1D*) of average cell width during one cell cycle is calculated using expression in *Ojkic et al. (2014)*. For a spherocylinder of pole-to-pole length $L$ and width $w$, the surface area is $S = wL\pi$, and volume is given by $V = \frac{\pi}{4}w^2L - \frac{\pi}{12}w^3$. In the case of *S. aureus*, surface area and volume are computed assuming prolate spheroidal shape using reported population averaged values of cell major axis, $c$, and minor axis $a$ (*Quach et al., 2016*). Surface area of a prolate spheroid is $S = 2\pi a^2 + \frac{2\pi ac^2}{\sqrt{c^2 - a^2}}\arcsin(\frac{\sqrt{c^2-a^2}}{c})$, and volume is $V = \frac{4\pi}{3}a^2c$.

## Cell growth simulations

We simulated the single-cell model using the coupled equations for the dynamics of cell length $L$, cell width $w$, and division protein production $P$ (*Figure 2C*). In simulations, when $P$ reaches the threshold $P_0 = \rho\pi w$, the mother cell divides into two daughter cells whose lengths are $0.5 \pm \delta$ of the mother cell. Parameter $\delta$ is picked from Gaussian distribution ($\mu = 0$, $\sigma = 0.05$).

For nutrient shift simulations we simulated $10^5$ asynchronous cells growing at a rate $k = 0.75\,\mathrm{h}^{-1}$ (*Figure 3*). In *Equation 5*, parameter $\beta = 4k/w$ is obtained from the fit to experimental data for $4k/w$ vs $k$ (*Figure 3—figure supplement 1A*) (*Si et al., 2017*), giving $\beta = 3.701k + 0.996$, where $k$ is in units of $\mathrm{h}^{-1}$, and $\beta$ in $\mathrm{h}^{-1}\,\mu\mathrm{m}^{-1}$. At t = 0 h we change $k$ corresponding to nutrient upshift ($k = 1.25, 2\,\mathrm{h}^{-1}$) or nutrient downshift ($k = 0.75, 0.25\,\mathrm{h}^{-1}$). We calculated population average of length and width (*Figure 3B*), and population average of aspect ratio of newborn cells (*Figure 3C*). Aspect ratio of newborn cells are binned in time and the bin average is calculated for a temporal bin size of 10 min. Examples of single cell traces during the nutrient shift are shown in *Figure 3—figure supplement 1B–E*.

FtsZ depletion experiment (*Zheng et al., 2016*) was simulated for $w = 1\,\mu m$ while $k_P$ was reduced to 40% of its initial value. This is consistent with the reduction of relative mRNA to ~40% corresponding to addition of 3 ng/ml of aTc to reduce *ftsZ* expression (*Zheng et al., 2016*). Our model predictions for the dependence of cell aspect ratio on $k_b/k_d$ is shown in *Figure 4—figure supplement 1B*.

Best fit for MreB depletion experiment (*Zheng et al., 2016*) was obtained for η ≈ 2.7, by simulating reduction in division protein production rate, $k_P$, and by varying $\beta$ so that width spans range from 0.9 to 1.8 µm. The best fit for long filamentous cells (resulting from DNA or cell-wall targeting antibiotics) was obtained for η ≈ 11.0 . Filamentation was simulated by decreasing $k_P$ and $\beta$ so that $w$ spans the range from 0.9 to 1.4 µm as experimentally observed (*Nonejuie et al., 2013*).

Open Source Physics (www.compadre.org) Java was used for executing the simulations and *Mathematica* 11 for data analysis, model fitting, and data presentation.

## Acknowledgements

We thank Suckjoon Jun lab (UCSD) for providing single cell shape data for *E. coli*, and Javier López-Garrido, Guillaume Charras, and Deb Sankar Banerjee for useful comments. SB gratefully acknowledges funding from EPSRC New Investigator Award EP/R029822/1, Royal Society Tata University Research Fellowship URF/R1/180187, and Royal Society grant RGF/EA/181044.

## Additional information

### Funding

| Funder | Grant reference number | Author |
| --- | --- | --- |
| Royal Society | URF/R1/180187 | Shiladitya Banerjee |
| Royal Society | RGF/EA/181044 | Shiladitya Banerjee |
| Engineering and Physical Sciences Research Council | EP/R029822/1 | Shiladitya Banerjee |

The funders had no role in study design, data collection and interpretation, or the decision to submit the work for publication.

## Author contributions
Nikola Ojkic, Conceptualization, Data curation, Formal analysis, Validation, Investigation, Visualization, Methodology, Writing—original draft; Diana Serbanescu, Data curation, Investigation, Methodology; Shiladitya Banerjee, Conceptualization, Resources, Supervision, Funding acquisition, Validation, Investigation, Visualization, Writing—original draft, Project administration

## Author ORCIDs
Shiladitya Banerjee (iD) https://orcid.org/0000-0001-8000-2556

## Decision letter and Author response
Decision letter https://doi.org/10.7554/eLife.47033.014
Author response https://doi.org/10.7554/eLife.47033.015

## Additional files
### Supplementary files
• Supplementary file 1. Sample size for collected bacterial shape data.
DOI: https://doi.org/10.7554/eLife.47033.010

• Supplementary file 2. Sample size for rod-shaped bacterial and archeal cell shapes.
DOI: https://doi.org/10.7554/eLife.47033.011

• Transparent reporting form
DOI: https://doi.org/10.7554/eLife.47033.012

### Data availability
All data generated or analysed during this study are referenced in the manuscript and supporting files.

The following datasets were generated:

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
