## [Decision Letter]

[Editors’ note: this article was originally rejected after discussions between the reviewers, but the authors were invited to resubmit after an appeal against the decision.]

Thank you for submitting your work entitled "Universal surface-to-volume scaling and aspect ratio homeostasis in rod-shaped bacteria" for consideration by *eLife*. Your article has been reviewed by two peer reviewers, and the evaluation has been overseen by a Reviewing Editor and a Senior Editor. The following individuals involved in review of your submission have agreed to reveal their identity: Charles W Wolgemuth (Reviewer #1).

The reviewers had no doubt that the subject matter of your paper is both interesting and timely, but both have raised questions about its significance, the nature of the underlying assumptions, and the level of "universality" that can be claimed. Our decision on your manuscript has been reached after consultation between the reviewers. Based on these discussions and the individual reviews below, we regret to inform you that your work will not be considered further for publication in *eLife*.

Reviewer #1:

This manuscript compiles data on the length, width, and growth rate of *E. coli* under a number of experimental perturbations, such as changes in growth medium, incubation in antibiotics, inhibited protein synthesis, etc. and shows that the surface area to volume ratio is strongly conserved. This result is consistent with recent work from Julie Theriot's group (which is cited in Harris and Theriot, 2016; 2018). In this manuscript, the author's also add data from other rod-shaped bacteria that show similar behavior (Figure 1G). The authors use this result to develop a model for aspect ratio regulation that is based on exponential growth of the bacterial length at fixed width, FtsZ production at a rate proportional to volume growth rate (which by assuming constant width also assumes that the FtsZ production is proportional to the length growth rate), and division that occurs when FtsZ production reaches a critical value that is proportional to the width. This model predicts a constant aspect ratio and the authors then go on to predict the dynamics under impulse type perturbations.

I have two main concerns with the manuscript.

First, the only novelty of the model is the assumption that there is a critical amount of FtsZ required to divide the cell and that this depends on the width. I think that this is a reasonable assumption, but I also feel that the overall results are fairly obvious. That is, it is not clear that the model provides a significant advance in our understanding. That said, there is also a small problem with the model, in that we would expect the binding rate of FtsZ to depend on the surface to volume ratio (which turns out not to matter, because the authors end up making assumptions that the rate the ring is built is equal to the protein production rate). Note also that there is a typo in the equation fordPrdt, which has *k*_d_ multiplying both rates.

Second, and more important, is that while the results match well with the data, there are a number of aspects of the presentation that are misleading. The title claims that the results/model presented here are universal. In Figure 1G, the authors select 7 bacteria to claim that the scaling of SA = 2 π V^(2/3)^ is ubiquitous among bacteria. As noted, this also suggests that an aspect ratio of ~4 is the rule for rod-shaped bacteria. This is not true. As an example, myxococcus xanthus has an aspect ratio around 7-8 and spirochetes have aspect ratios of ~30! Even more importantly, single species don't always maintain the same aspect ratio. In *B. subtilis*, the aspect ratio can vary between at least 3.8 – 8 (see Ilkanaiv et al., 2017). Therefore, this model may be applicable to *E. coli* (and possibly some other bacteria), but it is not universal.

Reviewer #2:

In this study the authors set out to study the size and shape of a wide range of 'rod-shaped' cells by collecting image data from at least seven different species and thousands to total conditions (genotype x nutrients x antibiotics). Across all these conditions, the authors find a simple scaling law to the surface area/volume ratio, namely a scaling that preserves the aspect ratio of the cell at roughly 4:1. Given this observation, they build a simple, mechanistically inspired, quantitative model for the growth of the cell. Using this model, they are able to tune three parameters (*k, k_p_*, β) to match a collection of genetic knockdown and antibiotic treatment experiments.

Understanding how cell size and shape homeostasis is maintained throughout the bacterial kingdom is a very interesting and important problem and these authors should be commended for pushing the community to consider that these mechanisms may be conserved across a wide phylogenetic range. However, given the extensive body of literature already available about cell size/shape homeostasis, and, in particular the review mentioned by the authors by Harris and Theriot, the scientific bar for productive engagement on this topic is already quite high. Much of the intellectual driving force for this work seems to follow directly from the hypothesis from Harris and Theriot that "While many studies have treated volume as the actively controlled parameter in this scenario, our recent work suggests that it is likely the other way around, and that SA/V is the actively regulated variable, with size following along as necessary [13]." The current work seeks to extend or provide alternatives for the mechanistic models presented in Harris and Theriot as well as integrate additional data in other species. However, given that the idea of SA/V scaling conservation is not new, appealing to a broad audience such as that of *eLife* would require experimental validation of their mechanistic model.

In addition to the core concerns around novelty of the central hypothesis and validity of the mechanistic model, there are a few issues the authors might choose to consider:

Major points:

1) The authors should clearly explain how their mechanistic model contrasts with the cell wall-focused model proposed by Harris and Theriot and should strive to propose experiments with predicted outcomes that would differentiate a peptidoglycan centric model from an FtsZ centric model. If the data already exist to rule out one of them, this should be clearly presented. As one such example, the authors show that tuning one parameter (*k_p_*) is consistent with the experimental notion of knocking down the production of FtsZ. However, they fail to show if there is quantitative agreement between the production rate of FtsZ and the amount they expect to need to change *k_p_*(40%).

2) The use of 'universal' in the paper's title significantly oversells of the breadth of species included in the observations and a power law describing data which span roughly one order of magnitude. While the authors do include a large collection of data, the collection is far from comprehensive for all size/shape data available and the authors do not clearly indicate why they limited themselves to the data they did. A quick literature search reveals anecdotal evidence of bacterial sizes that are much smaller than a micron such as *Brevundimonas* (PDA J Pharm Sci Technol. 2002 Mar-Apr;56(2):99-108.) to nearly a millimeter in length *Epulopiscium* (J. Protozoal., 35(4), 1988, pp. 565-569). Granted, these publications may not have the same type of data necessary to integrate it directly into their model, but for a discussion of the 'universal scaling', the authors should push themselves to cover as large of a length-scale as possible. When choosing a set of species for inclusion in this study, it seems like the microbiology community may have already picked an aspect ratio of about 4:1 in its definition of rod-shaped bacteria. For example, cells that have a much shorter aspect ratio are given the term ovoid or lancet (*Streptococcus pneumoniae*) or spherical (*Staphylococcus aureus* included here) and ones that are much longer are called filamentous (*Sphaerotilus natans*). Confusingly, these authors do not include species that have been traditionally classified as rod-shaped cells with a longer aspect ratio such as (*Helicobacter, Spiroplasma*, Spirochetes, Myxobacter).

3) I'm not entirely convinced that the universal scaling applies within the single cell data (Figure 1D). By plotting the single cell data from a variety of experiments, the range of the data seems to put a larger priority on the averages. However, within each condition there seems to be clear deviations from the 'single aspect ratio', consistent with the author's single cell growth model that cells grow without changing their diameter before dividing. This should result in a roughly factor of two change in aspect ratio from birth to division. I think this is what the authors refer to in the fourth paragraph of the Introduction but should discuss more fully.

4) I do not understand Figure 2B at all. In particular, the binning of the data that I have been able to find in Taheri-Araghi et al., 2015, is binned by the size of cells at birth, not the individual cell growth rate. Further, the authors do not describe how they go from the data in Taheri-Araghi et al., 2015, to the data in Figure 2B, but it could be that they obtained the raw data from the authors and performed a new type of analysis. If so, a description of this process should be included.

5) I am unclear about why the MreB and FtsZ knockdown data from Si et al. is included in the bulk Figure 1A data but the MreB and FtsZ knockdown data from Zheng et al. is treated as a completely separate experiment. If the approach that these two studies used was different, it may be helpful to explain why some data are included in one place and others are not.

---

## [Author Response]

[Editors’ note: the author responses to the first round of peer review follow.]

The mechanistic or molecular origin of bacterial aspect ratio control has remained an unsolved problem for more than four decades (see e.g. Zaritsky,2015; Zaritsky, 1975). In our manuscript we provide the first biophysical model for aspect-ratio homeostasis in rod-like bacteria and elucidate the underlying molecular mechanism that will inform future experimental studies. Our findings push the field of ‘bacterial cell size control’ to a new direction, which has so far focused on the individual control of cell volume, length or width, neglecting how bacterial length and width are coupled to give rise to rod-like cell shapes.

To support our model, we collected a large number of cell shape data (~5000 conditions) from many different laboratories, which indeed confirm that aspect ratio is conserved in *E. coli* (and 7 other organisms) under many different perturbations to the growth conditions (Figure 1). Importantly, our model also predicts under what conditions *E. coli* cells may deviate from their homeostatic aspect ratio of 4:1, and we tested our quantitative predictions for filamentous and spherical cell shapes against experimental data (Figure 2). Therefore, our thesis is not solely about the maintenance of 4:1 aspect ratio in *E. coli*, but more broadly about the control of bacterial cell shapes under many different perturbations.

It is evident from the reviewer comments that the principal issues with our manuscript lie in the presentation of results (e.g. claim of ‘universality’), and concerns about the novelty of our model in the context of previous studies. This is in part due to inadequate communication on our part. Having carefully read and deliberated on the reviewer comments, we believe that an improvement in the presentation of our results, an increased clarity of writing, and expanded description of the model will address all reviewer comments thoroughly and completely.

Reviewer #1:This manuscript compiles data on the length, width, and growth rate of E. coli under a number of experimental perturbations, such as changes in growth medium, incubation in antibiotics, inhibited protein synthesis, etc. and shows that the surface area to volume ratio is strongly conserved. This result is consistent with recent work from Julie Theriot's group (which is cited in Harris and Theriot, 2016; 2018). In this manuscript, the author's also add data from other rod-shaped bacteria that show similar behavior (Figure 1G). The authors use this result to develop a model for aspect ratio regulation that is based on exponential growth of the bacterial length at fixed width, FtsZ production at a rate proportional to volume growth rate (which by assuming constant width also assumes that the FtsZ production is proportional to the length growth rate), and division that occurs when FtsZ production reaches a critical value that is proportional to the width. This model predicts a constant aspect ratio and the authors then go on to predict the dynamics under impulse type perturbations.

We appreciate the succinct summary of our work. While our findings are consistent with a recent model proposed by Julie Theriot’s group, it is important to note the key difference: Harris and Theriot showed that rod-shaped bacteria (*E. coli*, *C. crescentus*) maintain a homeostatic surface-to-volume ratio (*S/V*) in a growth-rate dependent manner. Here we uncover a much stronger geometric constraint that bacteria (especially *E. coli*) maintain the relationship *S*=*γV*^2/3^ (with a constant *γ*), independent of growth rate. Furthermore Harris and Theriot model does not lead to aspect ratio control, as addressed in response to a comment from Reviewer #2.

I have two main concerns with the manuscript.First, the only novelty of the model is the assumption that there is a critical amount of FtsZ required to divide the cell and that this depends on the width. I think that this is a reasonable assumption, but I also feel that the overall results are fairly obvious. That is, it is not clear that the model provides a significant advance in our understanding.

Our model and analysis expand the current state of understanding in many ways:

– We provide the first biophysical model for aspect ratio control in bacteria and identify the molecular origins. We support our model by large population measurements (~5011 growth conditions in *E. coli*, 50 different bacterial species) and single cell measurements in mother machine (*n*~80,000). Why *E. coli* cells maintain a constant aspect ratio has been puzzling for more than half a century (Zaritsky, 1975; Zaritsky, 2015), with no prior existing model.

– The model for aspect ratio homeostasis provides a conceptual leap in the field of bacterial cell size control, by showing that added cell length for rod-like cells is coupled to their diameter. Previous phenomenological models treated cell length and diameter as independent control variables (Taheri-Araghi et al., 2015, Harris and Theriot, 2016).

– We think it’s not obvious that *E. coli* cells preserve their aspect ratios under multitude of perturbations to the nutrient conditions, ribosomes, protein overexpression or deletion (Figure 1A). Our model not only identifies that the maintenance of aspect ratio emerges from balanced biosynthesis of growth and division proteins (*k*/*k*_P_ constant), but also predicts under what perturbations cells may deviate from their homeostatic aspect ratio of 4:1 (Figure 4). We provide a quantitative, experimentally testable model for cell shape control that goes beyond just the regulation of FtsZ kinetics.

That said, there is also a small problem with the model, in that we would expect the binding rate of FtsZ to depend on the surface to volume ratio (which turns out not to matter, because the authors end up making assumptions that the rate the ring is built is equal to the protein production rate).

It is unclear why the FtsZ binding rate should depend on *S/V*. The rate equations are formulated in terms of the amount of the surface-bound and cytoplasmic FtsZ, and not their concentrations. If only the rate equations were formulated in terms of the concentration of cytoplasmic (*c*) and surface-bound proteins (*c_r_*), then the rate of increase in surface bound concentration of FtsZ would naturally depend on *S/V*.

dcrdt=kcr+kbVSc-kdcr

Second, if we were to consider changes in FtsZ binding rate with S/V, our simulations show that this has negligible effect on aspect ratio. For rod-shaped cells, *S/V* ~1/*w* in the first approximation, where *w* is cell width. Since during one cell cycle width doesn’t change, *S/V* stays approximately constant (new Figure 4—figure supplement 1A). If the width of the bacteria is changing due to changes in growth conditions, overall binding rate may be affected by (*S/V*) since the area of the Z-ring = δ*w* ~δ/(*S/V*), where δ is the lateral width of the FtsZ ring. *E. coli* width changes in different growth conditions from approximately 0.5 till 1 μm (Taheri-Araghi et al., 2015), so *S/V* can change by a maximum factor of 2. To address the effect of changes in binding rate, we simulated our dynamic model by changing the ratio *k*_b_/*k*_d_ across 4 order of magnitude. The figure in Figure 4-figure supplement 1B shows the dependence of cellular newborn aspect ratio (*n* = 10000, during steady-state growth) on *k*_b_/*k*_d_. In the limit *k*_b_>>*k*_d_, aspect ratio~4 as expected. However, the factor of 2 change even for the border line case of *k*_b_/*k*_d_ = 10, has negligible impact on cell aspect ratio.

However, as noted by the reviewer, the rate of FtsZ recruitment to the Z-ring (~10s, Soderstrom et al., Nat Commun 2018) is much faster than the growth rate. As a result, the rate at which the ring is built is determined by the production rate of FtsZ in the cytoplasm.

*Note also that there is a typo in the equation for*dPrdt*, which has kd multiplying both rates.*

We corrected the typo in the manuscript and thank the reviewer for pointing this out.

Second, and more important, is that while the results match well with the data, there are a number of aspects of the presentation that are misleading. The title claims that the results/model presented here are universal. In Figure 1G, the authors select 7 bacteria to claim that the scaling of SA = 2 π V^(2/3)^ is ubiquitous among bacteria. As noted, this also suggests that an aspect ratio of ~4 is the rule for rod-shaped bacteria. This is not true. As an example, myxococcus xanthus has an aspect ratio around 7-8 and spirochetes have aspect ratios of ~30! Even more importantly, single species don't always maintain the same aspect ratio. In B. subtilis, the aspect ratio can vary between at least 3.8 – 8 (see Ilkanaiv et al., 2017). Therefore, this model may be applicable to E. coli (and possibly some other bacteria), but it is not universal.

We apologize for the misunderstanding, which may have been triggered by a lack of clarity in our presentation. In our original submission, we did not claim that the 4:1 aspect ratio, or equivalently *S* = 2π*V*
^2/3^, is universal. Instead, we found that a ‘universal’ scaling law *S*=γ*V*^2/3^ is conserved amongst rod-shaped or coccoid bacterial species, implying the maintenance of a fixed aspect ratio (Figure 1A and E, dataset expanded). It is indeed possible for different bacteria to have different values for γ. For instance, in Figure 1E (previously 1G) we show that the coccoid *S. aureus* under different perturbations maintain the relation *S* = 4.92 *V*
^2/3^, implying preservation of the same scaling factor (2/3) while maintaining a different aspect ratio (1.38 +/- 0.18). In the same figure (1E), we now show data for a total of 48 different rod-shaped bacteria, and 1 rod-shaped Archaea (*H. vulcanii*), all of which remarkably follow the curve *S* = 2π*V*^2/3^.

Furthermore, our model also predicts how the aspect ratio and cell width can be altered by changing (*k*/*k*_p_) and (*k*/*β*), leading to filamentous or spherical cells, in agreement with available experimental data. In Figure 4 (previously Figure 2) we show that our model indeed predicts the breakdown of the 4:1 aspect ratio in *E. coli* under FtsZ or MreB perturbations.

However, the reviewer has made an excellent point that long filamentous cells, such as Spirochetes, do not necessarily conserve their aspect ratios. In Figure 1E, we now also include the data for Spirochetes, as one of the exceptions to the rule *S*=γ*V*^2/3^. We have therefore removed the term ‘universal’ from the title and Abstract of our paper. The fact, however, remains that *E. coli* remarkably conserve their aspect ratios under diverse size perturbations spanning two orders of magnitude (Figure 1A), and so do 50 other cell types (Figure 1E).

Motivated by the comments of reviewers 1 and 2, we now include a schematic in Figure 1D to illustrate the expected scaling relations for different bacterial shapes. Filamentous cells (Helicobacter, Spiroplasma, Spirochetes, Myxobacter) would likely follow the relation Sμ*V*, whereas coccoid or the rod-shaped cells follow the scaling law: *SμV*^2/3^.

Reviewer #2:In this study the authors set out to study the size and shape of a wide range of 'rod-shaped' cells by collecting image data from at least seven different species and thousands to total conditions (genotype x nutrients x antibiotics). Across all these conditions, the authors find a simple scaling law to the surface area/volume ratio, namely a scaling that preserves the aspect ratio of the cell at roughly 4:1. Given this observation, they build a simple, mechanistically inspired, quantitative model for the growth of the cell. Using this model, they are able to tune three parameters (k, k_p_, β) to match a collection of genetic knockdown and antibiotic treatment experiments.Understanding how cell size and shape homeostasis is maintained throughout the bacterial kingdom is a very interesting and important problem and these authors should be commended for pushing the community to consider that these mechanisms may be conserved across a wide phylogenetic range. However, given the extensive body of literature already available about cell size/shape homeostasis, and, in particular the review mentioned by the authors by Harris and Theriot, the scientific bar for productive engagement on this topic is already quite high. Much of the intellectual driving force for this work seems to follow directly from the hypothesis from Harris and Theriot that "While many studies have treated volume as the actively controlled parameter in this scenario, our recent work suggests that it is likely the other way around, and that SA/V is the actively regulated variable, with size following along as necessary [13]." The current work seeks to extend or provide alternatives for the mechanistic models presented in Harris and Theriot as well as integrate additional data in other species. However, given that the idea of SA/V scaling conservation is not new, appealing to a broad audience such as that of eLife would require experimental validation of their mechanistic model.In addition to the core concerns around novelty of the central hypothesis and validity of the mechanistic model, there are a few issues the authors might choose to consider:

We thank the reviewer for summarising the key aspects of our work and recognizing the importance of the field of study. Below we address some of the key comments raised above. “extensive body of literature already available about cell size/shape homeostasis” –A lot of work has been done over the past five years on developing phenomenological models for cell size control. Phenomenological models for the homeostasis of bacterial cell shape are treated the control of cell length separately from the control of cell width in rod-shaped bacteria. We provide a molecular model to show for the first time that bacterial cell dimensions are coupled to preserve aspect ratio, thereby linking the field of cell size and shape homeostasis.

“Much of the intellectual driving force for this work seems to follow directly from the hypothesis from Harris and Theriot” – Our model draws evidence from multiple recent experimental studies, while questioning the Harris and Theriot (HT) hypothesis. It is important to recognize the key distinctions between the two models. HT model does not lead to conservation of *S-to-V* scaling or aspect ratio, instead it leads to a model for the control of cell width (Eq. 3). HT model infers that *S/V* ratio is a function of growth media, such that cells reach a new homeostatic value of *S/V* upon perturbations in the growth rate. Here instead we propose a much stronger constraint that cells preserve the scaling relation, *S* = *μV*
^2/3^ (*μ* a constant) under diverse growth perturbations (~5000 conditions) across ~50 different bacterial species. Furthermore, HT model is agnostic about molecular mechanisms. Here we provide an explicit molecular candidate (FtsZ) for bacterial shape control, in agreement with exciting new evidence from Si et al., 2019. Taken together, our model integrates the adder model for cell size homeostasis with the regulation of *S/V* ratio and FtsZ, providing an integrative framework that successfully predicts bacterial shape control with only three physiological parameters.

“the idea of SA/V scaling conservation is not new” – We are not aware of any other studies that propose conservation of the scaling relation *S = μV*
^2/3^ across growth conditions, nor provide a model for it. Others have only shown evidence for the regulation of surface-to volume ratio by growth rate, which is a natural consequence of our model (Figure 1C).

“appealing to a broad audience such as that of *eLife* would require experimental validation of their mechanistic model” – Our model is tightly grounded in experimental data (see Figures 1-4), and we compare our model predictions extensively to experimental data, throughout the manuscript. As we are not an experimental lab, we have compiled data from a number of different laboratories to show that our model is consistent with all the available cell shape data across ~50 bacterial species and ~5000 growth conditions for *E. coli*. We definitely welcome suggestions to test our model further.

Major points:1) The authors should clearly explain how their mechanistic model contrasts with the cell wall-focused model proposed by Harris and Theriot and should strive to propose experiments with predicted outcomes that would differentiate a peptidoglycan centric model from an FtsZ centric model. If the data already exist to rule out one of them, this should be clearly presented.

We agree with the reviewer that clearer discussion of the contrast between our model and that of Harris/Theriot should be articulated in the manuscript. In the revised manuscript, we have expanded the discussion to highlight the key differences between these two models.

Foremost amongst the comparison is that Harris and Theriot proposes a homeostatic regulation of S/V in a growth-rate dependent manner. Whereas we propose a much stronger geometric constraint that the scaling relation *S* = *μV*
^2/3^ is preserved independent of growth rate. This result, however, does not contradict the model of Harris/Theriot.

Second, Harris and Theriot proposed a model where cells divide once a threshold amount of excess surface area material, Δ*A*, is accumulated in the cell. From this model it follows that, Δ*A* = Δ*V* (*β/k* – 2/*r*) = constant, where r is the cell radius of cross-section. This in turn 1, which is in contradiction to experimental data (Figure 1).

Third, we can indeed propose several experimental tests for our model, as highlighted in the revised manuscript:

– Our model would predict that FtsZ overexpression leads to minicells while FtsZ deletion would induce elongated phenotypes (Figure 4A). These predictions are consistent with data from Potluri et al., 1999, and Zheng et al., 2016.

– Oscillations in FtsZ amount would lead to cell size oscillations, in agreement with new data from Si et al., 2019.

– Total abundance of FtsZ scales with cell diameter, in agreement with data from Shi et al., 2017.

– We further predict that FtsZ knockdown would break aspect ratio preservation, whereas targeting cell wall precursors would change growth rate, but not alter aspect ratio or the scaling relation *S* = μ*V*
^2/3^. Figure 4—figure supplement 1C shows surface-to-volume scaling for *E. coli* cells treated with Fosfomycin that target MurA (affecting cell wall biogenesis) and FtsZ depletion. We find that Fosfomycin treated cells preserve the S~V^2/3^ scaling, whereas FtsZ depletion breaks the S~V^2/3^ scaling. This is a clear contrast between the role of cell wall precursors and FtsZ on cell shape control, implying that a cell wall precursor-based model alone is not sufficient to account for shape changes.

As one such example, the authors show that tuning one parameter (k_p_) is consistent with the experimental notion of knocking down the production of FtsZ. However, they fail to show if there is quantitative agreement between the production rate of FtsZ and the amount they expect to need to change k_p_ (40%).

Our model predicts that a reduction in FtsZ production rate to 40% of WT leads to observed phenotype in Zheng et al., 2016. This is consistent with reduction of relative mRNA to ~ 40% corresponding to addition of 3 ng/ml of aTc (Figure 2B of the Zheng et al.). We comment on this in our manuscript and thank the reviewer for pointing this out.

2) The use of 'universal' in the paper's title significantly oversells of the breadth of species included in the observations and a power law describing data which span roughly one order of magnitude. While the authors do include a large collection of data, the collection is far from comprehensive for all size/shape data available and the authors do not clearly indicate why they limited themselves to the data they did. A quick literature search reveals anecdotal evidence of bacterial sizes that are much smaller than a micron such as Brevundimonas (PDA J Pharm Sci Technol. 2002 Mar-Apr;56(2):99-108.) to nearly a millimeter in length Epulopiscium (J. Protozoal., 35(4), 1988, pp. 565-569). Granted, these publications may not have the same type of data necessary to integrate it directly into their model, but for a discussion of the 'universal scaling', the authors should push themselves to cover as large of a length-scale as possible. When choosing a set of species for inclusion in this study, it seems like the microbiology community may have already picked an aspect ratio of about 4:1 in its definition of rod-shaped bacteria. For example, cells that have a much shorter aspect ratio are given the term ovoid or lancet (Streptococcus pneumoniae) or spherical (Staphylococcus aureus included here) and ones that are much longer are called filamentous (Sphaerotilus natans). Confusingly, these authors do not include species that have been traditionally classified as rod-shaped cells with a longer aspect ratio such as (Helicobacter, Spiroplasma, Spirochetes, Myxobacter).

We address this point in response to the first reviewer. Both the reviewers have raised a pertinent point that long filamentous cells, such as Spirochetes, do not necessarily conserve their aspect ratios. In Figure 1E, we now include available shape data for Spirochetes, as one of the exceptions to the rule *S*=γ*V*
^2/3^. We have therefore removed the term ‘universal’ from the title and Abstract of our paper. The fact, however, remains that rod-shaped bacteria (*E. coli*) remarkably conserve their aspect ratios under diverse size perturbations spanning two orders of magnitude (Figure 1A).

In Figure 1E we have now expanded the dataset to cover two orders of magnitude by including 49 different rod-shaped bacterial species and 1 rod-shaped Archea. All of them lie on the curve *S*=γ*V*
^2/3^, confirming our predictions. In addition, we have also expanded the *E. coli* dataset by 30 more nutrient growth conditions (Gray et al., 2019), confirming our initial statement of aspect-ratio homeostasis.

We are grateful to the reviewer for providing the papers reporting drastic volume range in bacteria spanning 2 orders of magnitude. Bacteria that we include in Figure 1E are those that are known divide using FtsZ machinery during binary fission. This is to maintain consistency with our model that is based on FtsZ regulation. For this reason, we did not include *Epulopiscium* in our analysis. We also did not include *Sphaerotilus natans* in our graph as we could not find good shape measurements for it. In line with reviewer’s comments we have now included longer filamentous cells in Figure 1E. We have also introduced a new cartoon in Figure 1D showing how long filamentous cells that keep their width constant, would have a different scaling law S~V.

3) I'm not entirely convinced that the universal scaling applies within the single cell data (Figure 1D). By plotting the single cell data from a variety of experiments, the range of the data seems to put a larger priority on the averages. However, within each condition there seems to be clear deviations from the 'single aspect ratio', consistent with the author's single cell growth model that cells grow without changing their diameter before dividing. This should result in a roughly factor of two change in aspect ratio from birth to division. I think this is what the authors refer to in the fourth paragraph of the Introduction but should discuss more fully.

In our original submission, we had already explored in detail the deviation from 2/3 scaling in the single-cell data (Figure 2—figure supplement 1A-B). The main reason for the deviation from 2/3 scaling comes from large fluctuations in newborn length for a given width of bacteria. Using our model, we can quantitatively explain the deviation from the universal scaling by incorporating experimentally measured fluctuations in cell width and length, in agreement with experimental data. We have now attempted to explain this better in the manuscript and in the supplementary figure caption.

4) I do not understand Figure 2B at all. In particular, the binning of the data that I have been able to find in Taheri-Araghi et al., 2015, is binned by the size of cells at birth, not the individual cell growth rate. Further, the authors do not describe how they go from the data in Taheri-Araghi et al., 2015, to the data in Figure 2B, but it could be that they obtained the raw data from the authors and performed a new type of analysis. If so, a description of this process should be included.

We were kindly provided with the raw data for single-cell width and length at various growth rates (conditions) by the Suckjoon Jun lab. We reanalysed the data, performed the necessary binning and analysis. We have clearly stated this in the Appendix and in each figure caption.

5) I am unclear about why the MreB and FtsZ knockdown data from Si et al. is included in the bulk Figure 1A data but the MreB and FtsZ knockdown data from Zheng et al. is treated as a completely separate experiment. If the approach that these two studies used was different, it may be helpful to explain why some data are included in one place and others are not.

For consistency, we now plot the MreB and FtsZ knockdown data from Si et al. in Figure 4B. The knockdown data from *Si* et al. cover a small dynamic range so it is hard to extract a clear trend from these data alone. This is presumably because cells in those knockdown experiments were grown in slow growth media (MOPS glucose + 6 a. a., with growth rate ~0.75 h-1) and small perturbations, whereas the data from Zheng et al. that show drastic cell shape changes (Figure 4B) are obtained from experiments on rich media (RDM + glucose, with growth rate 1.6 h-1) and large perturbations. The trend in Si et al. seems to be consistent with those in Zheng et al.